# Controllable Synthesis, Formation Mechanism, and Photocatalytic Activity of Tellurium with Various Nanostructures

**DOI:** 10.3390/mi15010001

**Published:** 2023-12-19

**Authors:** Huan Wang, Hanlin Zou, Chao Wang, Sa Lv, Yujie Jin, Hongliang Hu, Xinwei Wang, Yaodan Chi, Xiaotian Yang

**Affiliations:** 1Key Laboratory for Comprehensive Energy Saving of Cold Regions Architecture of Ministry of Education, Jilin Jianzhu University, Changchun 130118, China; wanghuan@jlju.edu.cn (H.W.); wangchao@jlju.edu.cn (C.W.); lvsa82@163.com (S.L.); chiyaodan@jlju.edu.cn (Y.C.); 2Department of Materials Science, Jilin Jianzhu University, Changchun 130118, China; 19819830942@163.com (H.Z.); jinyujie@jlju.edu.cn (Y.J.); huhongliang@126.com (H.H.); 3Engineering Research Center of Optoelectronic Functional Materials, Ministry of Education, School of Materials Science and Engineering, Changchun University of Science and Technology, Changchun 130022, China; 4Department of Chemistry, Jilin Normal University, Siping 136000, China

**Keywords:** tellurium, various nanostructures, formation mechanism, photodegradation, photocatalytic H_2_ production

## Abstract

Telluriums (Te) with various nanostructures, including particles, wires, and sheets, are controllably synthesized by adjusting the content of polyvinylpyrrolidone (PVP) in a facile solvothermal reaction. Te nanostructures all have complete grain sizes with excellent crystallinity and mesopore structures. Further, the formation mechanisms of Te nanostructures are proposed to be that the primary nuclei of Te are released from the reduction of TeO_3_^2−^ using N_2_H_4_·H_2_O, and then grow into various nanostructures depending on the different content of PVP. These nanostructures of Te all exhibit the photocatalytic activities for the degradation of MB and H_2_ production under visible light irradiation, especially Te nanosheets, which have the highest efficiencies of degradation (99.8%) and mineralization (65.5%) at 120 min. In addition, compared with pure Te nanosheets, the rate of H_2_ production increases from 412 to 795 μmol∙h^−1^∙g^−1^ after the introduction of Pt, which increases the output by nearly two times. The above investigations indicate that Te with various nanostructures is a potential photocatalyst in the field of degradation of organic pollutants and H_2_ fuel cells.

## 1. Introduction

In recent decades, the energy crisis and environmental pollution have aroused widespread concern due to increases in energy consumption and human activities [1,2,3]. With increasingly serious environmental pollution, the demand for the treatment of industrial wastewater with toxic pollutants is increasing, especially for wastewater containing Rhodamine B (RhB) [4], Methylene blue [5], phenol [6], tetracycline chloride [7], ciprofloxacin [8,9] and its by-products. Meanwhile, hydrogen (H_2_) can be considered an ideal energy source in the future due to its high energy capacity and non-polluting nature [10,11]. Therefore, it is an urgent need to develop an integrated technology for pollutant elimination and hydrogen production. So far, many photocatalysts have been reported, including TiO_2_, ZnO, NiO, WO_3_, CdS, g-C_3_N_4_, MoS_2_, etc. [12,13,14,15,16,17,18,19]; however, most of these photocatalysts are only active under ultraviolet light, seriously limiting the utilization efficiency of solar energy. Therefore, it is of great significance to develop low-cost, efficient, and stable photocatalysts combined with the degradation of organic pollutants and hydrogen production under visible excitation.

Tellurium (Te) is a P-type semiconductor with a narrow band gap energy of 0.35 eV. Its hexagonal crystal structure has six spiral chains at each corner and one in the center, which are held together by van der Waals forces [20]. Te nanomaterials have intriguing features such as superior carrier mobility, strong light absorption capacity, high ductility, and good environmental stability due to their peculiar chiral-chain crystal lattice structure [21,22]. Te with various nanostructures has been attracting more attention in many fields, due to its unique properties, such as higher photoconductivity [23,24], nonlinear optical response [25] and efficient photocatalysis [26], etc. Zero-dimensional (0D) Te nanoparticles with smaller sizes are synthesized via treatment using laser ablation and ultrasonic exfoliation, applied in photovoltaic devices [27] and the photocatalytic degradation of rhodamine B (RhB) [28]. Highly ordered 1D Te nanowires possessing photoconductivity and the field emission property are also prepared via the catalyst-free physical vapor deposition method, realizing the application in flat panel displays [29]. In addition, 2D Te nanosheets with high hole mobility are successfully obtained using liquid-phase exfoliation; these nanosheets have high photoconductivity and a nonlinear optical response to be used in all-solid-state fiber lasers [30]. Based on the above studies, it is found that researchers have focused almost exclusively on the synthesis and application of Te with a single nanostructure. However, until now, there are very few literature reports on the controllable synthesis, formation mechanism, and photocatalytic degradation activity of Te with various nanostructures.

Herein, Te with various nanostructures is controllably synthesized by adjusting the content of PVP as molecular templates in the solvothermal reaction. Te nanostructures all have complete grain sizes with excellent crystallinity and mesopore structures. In addition, the formation mechanisms of Te nanostructures are proposed to include two steps of nucleation and growth; namely, the primary nuclei of Te are released from the reduction of TeO_3_^2−^ using N_2_H_4_·H_2_O, and then grow into various nanostructures from the different contents of PVP. These Te nanostructures all possess degradation activities for organic methylene blue molecules (MB) under visible light irradiation, in which Te nanosheets show the best photocatalytic degradation activity, corresponding to the highest degradation efficiency (99.8%) and excellent mineralization efficiencies (65.5%) at 120 min due to their largest specific surface areas and crystalline surface. In addition, the nanosheets also showed excellent hydrogen production rates (412 μmol∙h^−1^∙g^−1^), especially following the introduction of Pt nanoparticles (795 μmol∙h^−1^∙g^−1^), which increased the yield by nearly two times. Based on the above results, Te with various nanostructures is expected to be an ideal candidate photocatalyst in the field of photocatalytic degradation and H_2_ production.

## 2. Experimental Process

### 2.1. Synthesis of Te with Various Nanostructures

All the chemicals were analytical pure without further purification. The experimental process was as follows: Initially, Na_2_TeO_3_ (0.46 g), NH_3_·H_2_O (16.6 mL), and N_2_H_4_·H_2_O (8.38 mL) were added to double-distilled water (160 mL) in a beaker, following the addition of different PVP contents (0.5 g, 1.0 g, and 1.5 g, respectively) to synthesize various nanostructures. Afterwards, the mixture solution was transferred into a Teflon-lined stainless-steel autoclave with a capacity of 200 mL, and then maintained at 180 °C for 4 h. After that, the mixture solution was cooled and centrifuged to obtain the solid materials, which were washed three times with distilled water and alcohol. The products were dried at 60 °C for 12 h to form silver gray samples via grinding. Te with various nanostructures was synthesized via a facile solvothermal method.

### 2.2. Characterizations of Te with Various Nanostructures

The morphologies of synthesized Te nanostructures were observed via an FEI Quanta 200 F Scanning Electron Microscopy (SEM). X-ray Diffraction (XRD) data were collected on a D8-Focus powder diffractometer made by the Bruker company in Germany. The functional groups of materials were analyzed via Fourier transform infrared spectroscopy (FTIR, Thermo Nicolet Nexus, using a KBr beam splitter). The Raman spectrum was measured using a high-resolution confocal Raman microscope (Horiba Lab Ram Hr Evolution) at room temperature with an excitation wavelength of 532 nm. The nitrogen sorption isotherms were measured at the temperature of liquid nitrogen (77 K) using a BELSORP-MINI analyzer with the samples being degassed at 100 °C for 3 h before analysis. The surface area and pore-size distribution curves were calculated using the Brunauer–Emmett–Teller (BET) and Barrett–Joyner–Halenda (BJH) method, respectively.

### 2.3. Photocatalytic Activity Test

#### 2.3.1. Photocatalytic Degradation of Methylene Blue (MB)

The photocatalytic activities were determined via the photodegradation of MB for as-prepared Te samples as photocatalysts under visible light using a 300 W xenon lamp with a cut-off filter of 420 nm, which was placed 5 cm away from the Pyrex reactor. The degradative process was maintained at 25 °C by circulating cooling water. 50 mg of Te samples was dispersed in MB aqueous solution (100 mL, 10 mg/L). Before the lamp was switched on, the adsorption–desorption equilibrium was reached after 15 min in the dark. The photocatalytic degradation of MB detection was carried out in 15 min intervals using a UV-vis spectrophotometer (UV-2450, Shimadzu, Kyoto, Japan), after which the 5 mL suspension was centrifuged to take away the Te sample powders. In addition,, to study their photocatalytic stability, the prepared Te nanosheets were cycled five times under the same reaction conditions.

#### 2.3.2. Photocatalytic Hydrogen (H_2_) Evolution

The reaction of photocatalytic H_2_ evolution for the Te samples with various nanostructures was carried out in a 500 mL Pyrex glass reactor. Powder catalysts (100 mg) were dispersed in a mixed solution of H_2_O (180 mL) and methanol (20 mL, as a sacrificial reagent). As a contrast experiment, H_2_PtCl_6_ was added to the above-mentioned mixed solution to obtain Pt (1 wt%) as a cocatalyst via a photoreduction reaction. After that, the degasification of the reactor was carried out using a pump. Meanwhile, the mixture was continuously magnetically stirred for 20 min (500 r/min) to obtain the equilibrium of adsorption and desorption before irradiation. A 300WXe lamp equipped with a UV cut-off filter (λ > 420 nm) (Beijing China Education Au-Light Co., Ltd., Beijing, China) was placed 5 cm above the photocatalytic reactor. To maintain the experimental temperature effectively, a water-cooling system was applied to the periphery of the photoreactor. The amount of H_2_ evolution from water splitting was monitored using a gas chromatograph thermal conductivity detector (GC 2060). In addition, high-purity N_2_ was used as a protective and operating gas in the reaction tests of photocatalytic H_2_ evolution.

### 2.4. Photoelectrochemical Measurements

Photoelectrochemical tests were performed using an electrochemical workstation (CHI852C) with a 300 W Xe-lamp with a cut-off filter (λ ≥ 420 nm). Typically, the process of making a working electrode is as follows: 0.1 g of Te samples was dispersed into the mixture of ethanol glycol (3.0 mL) and oleic acid (0.03 mL) along with ultrasonic stirring for 30 min. Afterwards, the above-mentioned mixed colloidal dispersion was dropped onto a 2 cm × 4 cm FTO glass substrate to be vacuum-dried for 10 h at 80 ºC. In addition, Platinum wire and Ag/AgCl were applied to the counter electrode and the reference electrode, respectively. The transient state photocurrent and electrochemical impedance spectroscopy (EIS) were tested using Na_2_SO_4_ (0.2 M) aqueous solution and KCl (0.1 M) solution as the electrolytes, respectively.

## 3. Results and Discussion

### 3.1. Morphological and Structural Characterization

Tellurium nanocrystals with various morphologies are prepared at 180 °C for 4 h, and disparate dimensions are obtained by adjusting the content of polyvinylpyrrolidone (PVP) during the solvothermal approach. The morphology and size of as-synthesized typical Te samples is characterized via SEM, as shown in Figure 1. Briefly, 0D nanoparticles with smaller sizes and good distribution are exhibited by adding 0.5 g of PVP (Figure 1b and Appendix A). In addition, we selected nearly 200 nanoparticles of Te via SEM, as shown in Figure 1b, and calculated their average particle size to be 92.7 nm through particle size statistics (see Appendix A). For comparison, without adding PVP, the sample shows large clusters of nanoparticles with uneven sizes (Figure 1a). With the addition of 1.0 g PVP, well-defined, straight nanowires with an average diameter of 25 nm are synthesized (Figure 1c). When increasing the PVP content to 1.5 g, a comparatively regular nanosheet of Te is presented with a uniform and smooth surface that shows a length and width of 5.2 μm × 3.3 μm; thus, the synthesized Te nanosheets are relatively large (Figure 1d). In addition, a large number of 2D Te nanosheets are presented in Appendix A, which they all have a uniform lamellar structure. Furthermore, the thickness of the Te nanosheet is measured using an atomic force microscope (AFM), and the region shown by a white line corresponds to a thickness of about 8.7 nm in Appendix A. Based on the changes in the crystallization and morphology of Te products, these results clarify that the morphology of Te nanocrystals is regulated by the PVP molecules.

The XRD patterns of Te with various nanostructures are presented in Figure 2a. Obviously, all diffraction peaks of Te products are narrow and strong, demonstrating better crystallinities, and are absolutely indexed to the hexagonal phase (JCPDS 36-1452). Comparatively, the intensities of diffraction peaks for (101) planes are the highest, indicating the preferred growth of Te along the c-axis. The structural information of Te is recorded in the Raman spectra, which consists of characteristic vibration modes of A_1_, E_1_, and E_2_ (Figure 2b). Therein, the A_1_ mode is attributed to the chain extension caused by the atoms moving on the basal plane [31], the E_1_ mode is mainly caused by bond bending due to the rotation of the a-axis, and asymmetric stretching along the c-axis results in the E_2_ mode [32]. In addition, obviously, the red shift is gradually shown for the A_1_ modes following the increase in PVP content, possibly attributed to the decrease in basal plane vibration frequency [33]. Further, the FTIR spectra are measured (Figure 2c) to reflect the interactions between PVP and Te. The main characteristic peaks at 1654 cm^−1^ also exhibit a red shift, corresponding to the stretching vibration of C=O in PVP, implying that Te atoms are absorbed on nitrogen atoms of five-membered heterocycles of PVP [34], which is consistent with the Raman spectra. Meanwhile, the increase in peak intensity suggests that various nanostructures of Te are probably formed as the PVP content is increased. In addition, the information on textural properties for Te nanostructures is provided by N_2_ adsorption –desorption isotherms, where the characteristics of type IV with clear H_3_ hysteresis are shown, confirming the existence of mesopore architectures (Figure 2d). The BET surface areas are 56, 84, and 93 m^2^g^−1^, corresponding to nanoparticles, nanowires, and nanosheets, respectively. The Te nanosheets have the largest specific surface areas. The above-mentioned results provide direct evidence for the elaboration of our formation mechanism.

### 3.2. Optical and Photoelectrochemical Properties

To further investigate their optical properties, UV-vis absorption spectra of Te samples were examined. Among them, 50 mg of Te nanostructure samples was dissolved in 200 mL of acetone solution for ultrasonic treatment for 30 min, and after standing for 10 min, the supernatants were taken for UV-vis absorption spectra tests. As shown in Figure 3, the obvious wide characteristic absorption peaks at around 680 nm are exhibited from 400 to 1000 nm for Te materials with various nanostructures, indicating that all of them can efficiently use the wavelengths of the full visible region to produce photo-generated carriers. Meanwhile, from the intensity change of the curve, it can be seen that the adsorption ability of Te samples gradually increases from 0D to 2D, which may be related to the nanostructure type. Notably, 2D Te nanosheets exhibit the strongest absorption peaks, mainly ascribed to their excellent light scattering ability [21]. The ability to fully absorb visible light implies that Te materials with various nanostructures have more significant research value in the application field of visible light catalysis.

To elucidate the separation and transportation of photoexcited charge carriers, a photoelectrochemical test was conducted as an effective method of evaluation. Figure 4a displays the photocurrent responses of the various nanostructures for Te that are all regular on/off cycles under visible light irradiation, in which the photocurrent density is proportional to their photocatalytic efficiencies in the photocatalytic reaction. Among them, the 2D Te nanosheets produce a higher photocurrent density, indicating that the larger amount of photogenerated electrons could be quickly transferred to the surface because of the higher orientation degree of the (101) facet [27].

To clarify this further, electrochemical impedance spectroscopy (EIS) was also employed. As illustrated in Figure 4b, the Nyquist plot diameters (namely, the arc semicircle) of these samples reflects their charge transfer resistances. Visibly, the diameters of arc semicircles are significantly increased from 0D to 2D for Te samples. As expected, the 2D Te nanosheets exhibited the smallest arc semicircle, implying that the 2D nanostructure with a higher (101)-oriented degree can effectively separate and transfer photogenerated charge carriers by shortening the transport pathway of charge carriers. The above research results indicate that the 2D nanostructures with a higher (101)-oriented degree of Te are more conducive to enhancing its photocatalytic activity.

### 3.3. The Formation Mechanisms of Various Nanostructures for Te

The formation mechanisms are interpreted for Te with various nanostructures (Figure 5). The formation processes are mainly divided into two parts: nucleation and growth. At the initial stage of the reaction, Te atoms are gradually released from the reduction of TeO_3_^2−^ using N_2_H_4_·H_2_O. Then, PVP–Te complexes are formed due to Te atoms being successfully adsorbed on the nitrogen atoms of five-membered heterocycles of PVP via electrostatic interaction [22,35], as shown in the following equations:TeO_3_^2−^ + N_2_H_4_·H_2_O → Te + N_2_ + H_2_O + 2OH^−^(1)
PVP + Te → PVP-Te complex(2)

After the nucleation, the various nanostructures of Te are formed in the ongoing reaction. When adding 0.5 g of PVP, a small amount of PVP content is coated on the surface of Te atoms to effectively inhibit the clusters, further forming 0D Te nanoparticles with smaller sizes and better dispersion, attributed to the prevention of high surface energies of metastable Te atoms through the repulsive force from the alkyl group of PVP [21,22]. Comparatively, without PVP, large volume clusters of Te atoms occur. When the PVP content is increased to 1.0 g, more nitrogen atoms of PVP are provided to absorb the Te atoms [21], possibly accelerating the 1D growth of Te along the long chain of PVP [36]. Perhaps, the Te atoms with two polar end faces grow along the polar c-axis to form the typical 1D nanostructure, due to the hexagonal structure and the anisotropy of Te atoms [20]. At a higher PVP content (1.5 g), most of the PVP molecules are coated onto the polar facets of Te atoms, which restrains growth along the c-axis and promotes the growth of the other two nonpolar facets under the thermodynamics, forming 2D Te nanosheets [22].

However, our understanding is limited for the formation mechanism. Therefore, more detailed studies will be carried out to elucidate the mechanism.

### 3.4. Photocatalytic Degradation of MB and Hydrogen Generation

#### 3.4.1. Photocatalytic Degradation of MB

The photocatalytic activities of Te with various nanostructures are further studied through the photodegradation of MB dye molecules in water under visible light irradiation. When magnetically stirred for 15 min, the concentration of MB solution appeared to reduce slightly using these nanostructured photocatalysts in the dark, suggesting that physical adsorption has a minimal effect on the reduction in dye concentration. Furthermore, the self-degradation of MB can be almost negligible in the blank experiment. However, the degradation efficiencies of MB are varied with the various nanostructures of Te. Comparatively, the clusters, the nanoparticles, the nanowires, and the nanosheets for Te presented different degradation efficiencies of 41.3%, 76.2%, 96.4%, and 99.8%, respectively, at 120 min (Figure 6a and Appendix A). Obviously, the degradation efficiency of Te nanosheets is significantly higher than those of the other nanostructures, ascribed to its 2D nanostructure with a larger specific surface area to make more coordination sites on the unsaturated surface for interaction with dye molecules, thus enabling the efficient transport of reactive species to enhance photocatalytic activity [15,29]. Corresponding to the above degradation efficiencies, photocatalytic kinetics are further provided for the degradation of MB (Figure 6b). The pseudo first-order kinetic equation for ln (C_0_/C_t_) = kt is used to reflect the photodegradable kinetics, in which k, t, C_0_, and C_t_, are the rate constant, irradiation time, and concentration of MB solution before and after irradiation [20]. Based on the above equation, the k values are calculated to be 0.018, 0.021, 0.032, and 0.068 min^−1^ for the various nanostructures of Te (see Appendix A). Te nanosheets have the highest k value (0.068 min^−1^), indicating the fastest degradation rate for MB solution, which is consistent with the above results.

The temporal evolution of the absorption peak of MB solution at 665 nm using the Te nanostructures is shown in Figure 6c and Appendix A. With the increase in the degradation time, the intensity of the absorption peak for MB is decreased gradually. In addition, the intensity of absorption disappeared completely when using Te nanosheets after 120 min of irradiation in Figure 6c. Appendix A also exhibits the mineralization behavior (TOC) of MB when using the various nanostructures of Te. The comparison results of TOC show that the mineralization efficiencies of MB are 13.5%, 25.8%, 53.4%, and 65.5%, corresponding to the nanoclusters, the nanoparticles, the nanowires, and the nanosheets for Te after 120 min of irradiation. However, no mineralization efficiency of MB was obtained for the nanoclusters of Te. The mineralization efficiency of MB using Te nanosheets is more than 2.5 times that of nanoparticles, which further indicates that Te nanosheets have excellent photodegradation activity.

To clarify the role of the main reactive species, three scavengers are applied in the photocatalytic degradation of MB for Te nanosheets, such as ethylene diamine tetraacetic acid (EDTA), iso-propanol (IPA), and benzoquinone (BQ), as scavengers of the quench hole (h+), hydroxyl radical (•OH) and superoxide radical (•O^2−^), respectively. As shown in Figure 6d, the degradation efficiency of MB is 99.2% in the absence of any scavengers after illumination for 1 h. When BQ was added to the suspension, the degradation efficiency of MB significantly decreased from 99.2% to 16.8%, indicating that •O^2−^ is the most active free radical in photodegradation.

#### 3.4.2. Photocatalytic Hydrogen (H_2_) Generation

Photocatalytic H_2_ production activities of Te samples with various morphologic structures were monitored via a water splitting reaction, in which methanol was chosen as the sacrificial reagent under visible light irradiation (λ ≥ 420 nm). The amount of H_2_ evolution for all the samples is shown for the 2.5 h of the water splitting process in Figure 7a. Clearly, Te nanosheets possess the best activity of H_2_ production in comparison with that of other samples. Correspondingly, the H_2_ production rate revealed the various degrees of photocatalytic activity for H_2_ evolution, as seen in Figure 7b. The average rates of H_2_ evolution are 0, 246, 302, and 412 μmol∙h^−1^∙g^−1^ for the clusters, the nanoparticles, nanowires, and nanosheets of Te, respectively (Figure 7b). Among them, Te nanosheets display the highest H_2_ production rate (412 μmol∙h^−1^∙g^−1^), which is nearly twice as much as that of Te nanoparticles. The result may mainly originate from the largest specific surface area offering more active points, and the strongest oriented degrees of the (101) facet promoting fast e-transfer along the direction of the facet for Te nanosheets [21,22]. In addition, Te clusters have almost no photocatalytic activity in H_2_ generation due to the accumulation of particles causing small specific surface areas and their amorphism going against producing photogenerated carriers [29,30,31]. As a comparison, after adding 1 wt% Pt as a cocatalyst, the significant increase rate in H2 activity was exhibited under the same experimental condition. Notably, with the introduction of Pt, the generated H_2_ rates are all increased significantly for the original samples, in which Te nanosheets exhibit a nearly two-fold increase from 412 to 795 μmol∙h^−1^∙g^−1^. The above comparison result suggests that the 2D nanosheets could be the most effective photogenerated e-transport channels, loading Pt onto the Te surface to enhance the carrier’s separation. Meanwhile, it will provide inspiration for designing new photocatalytic materials with semiconductor heterostructures based on Te as upholders.

#### 3.4.3. Photocatalytic Recyclability and Stability

The recyclability of the photocatalyst has an important impact on the realization of its future reuse. Therefore, Te nanosheets as a representative are used to carry out the cycle tests of photocatalytic degradation of MB and photocatalytic H_2_ production. Obviously, in Figure 8, the efficiencies of MB degradation and hydrogen production are almost not significantly decreased after five cycles, indicating that the Te nanosheets can be reused repeatedly as stable photocatalysts. In addition, as shown in Appendix A, no substantial changes for the structure of Te nanosheets are observed before or after usage, indicating that the Te nanosheet photocatalyst was remarkably stable.

## 4. Conclusions

Te with various nanostructures is controllably synthesized via adjusting the content of PVP as a molecular template via a facile solvothermal method. Te nanostructures all show clear grain boundaries and excellent crystallinity. In addition, these Te nanomaterials form mesoporous structures. Possible formation mechanisms are proposed that divide the formation processes into two steps, namely, nucleation and then the growth of nanostructures. These nanostructures of Te all exhibit the photocatalytic activities for the degradation of MB and H_2_ production under visible light irradiation, especially Te nanosheets, which have the highest efficiencies of degradation (99.8%) and mineralization (65.5%) at 120 min. In addition, compared with pure Te nanosheets, the rate of H_2_ production increased from 412 to 795 μmol∙h^−1^∙g^−1^ after the introduction of Pt, which increased the output by nearly two times. Based on the results, Te with various nanostructures can be a potential photocatalyst, and is expected to have further applications in the field of photocatalysis for the photodegradation of organic pollutants and photocatalytic hydrogen fuel cells.

## Figures and Tables

**Figure 1 micromachines-15-00001-f001:**
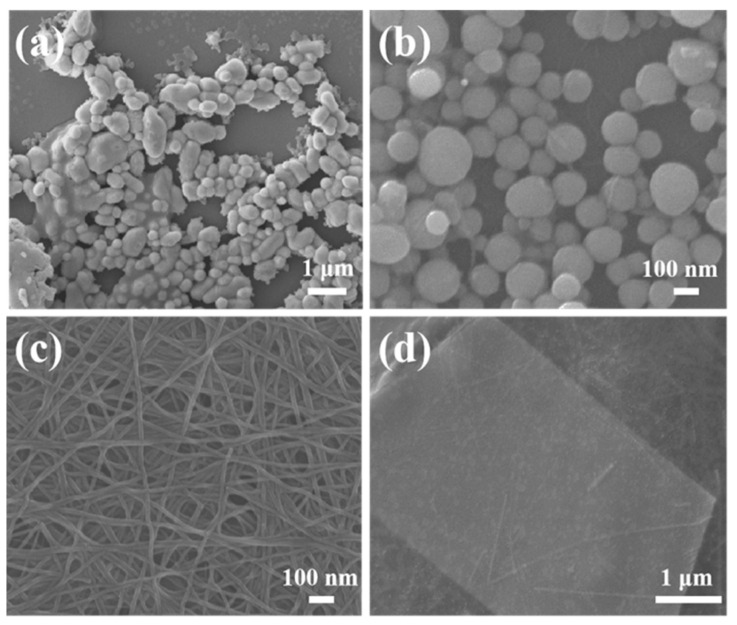
SEM images of as-synthesized Te with various nanostructures at the different contents of PVP: (**a**) 0 g, (**b**) 0.5 g, (**c**) 1.0 g, and (**d**) 1.5 g.

**Figure 2 micromachines-15-00001-f002:**
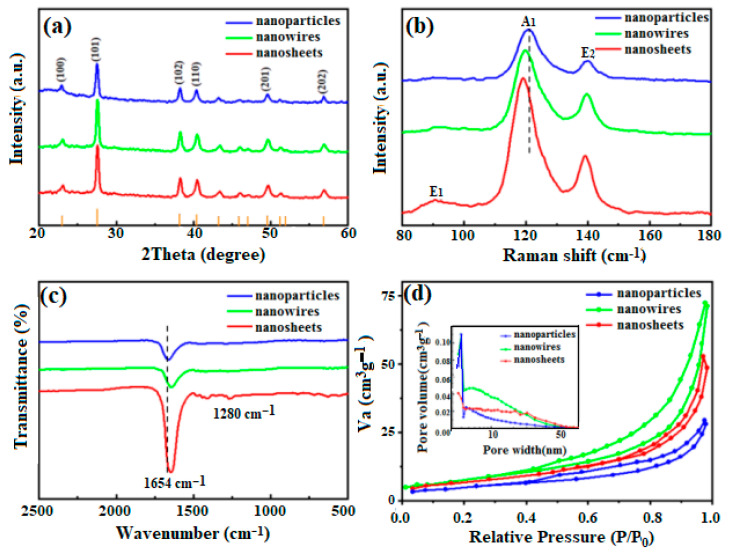
(**a**) XRD patterns, (**b**) Raman spectra, (**c**) FTIR spectra, and (**d**) nitrogen adsorption–desorption isotherms (pore size distribution plots, inset) of Te with various nanostructures.

**Figure 3 micromachines-15-00001-f003:**
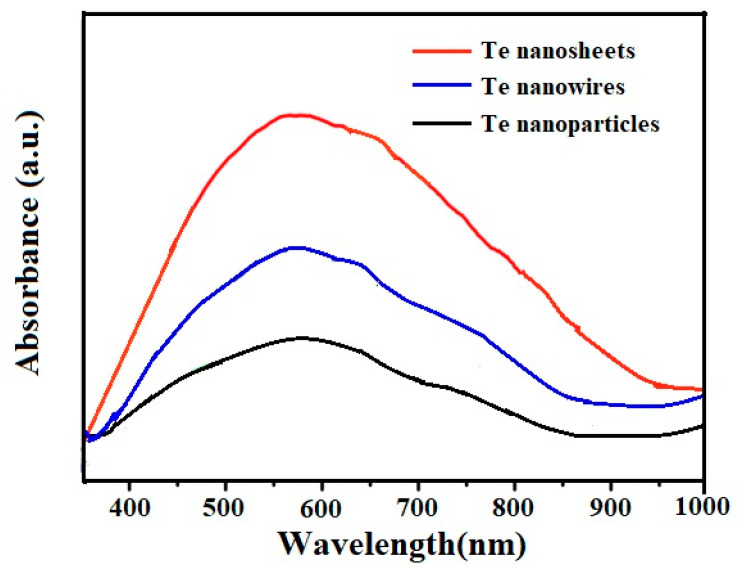
UV-vis absorption spectra for Te with various nanostructures.

**Figure 4 micromachines-15-00001-f004:**
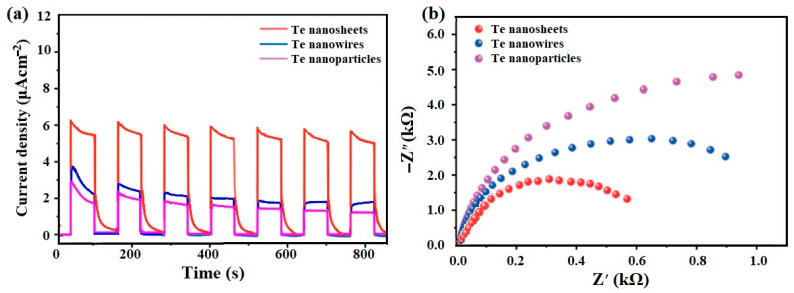
(**a**) Photocurrent responses and (**b**) EIS spectra of Te with various nanostructures.

**Figure 5 micromachines-15-00001-f005:**
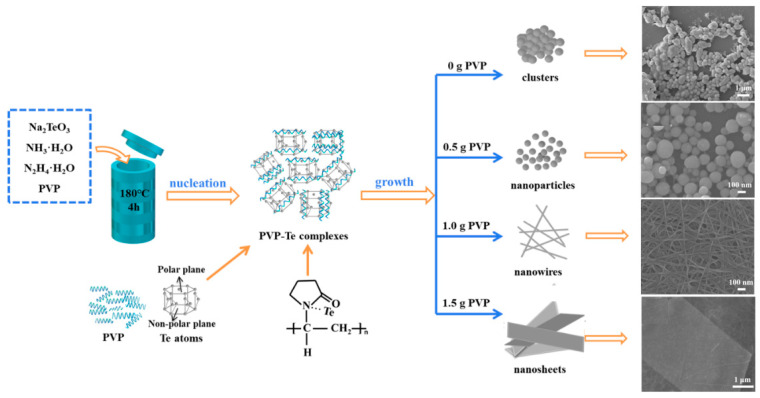
The schematic of the formation mechanism of Te with various nanostructures.

**Figure 6 micromachines-15-00001-f006:**
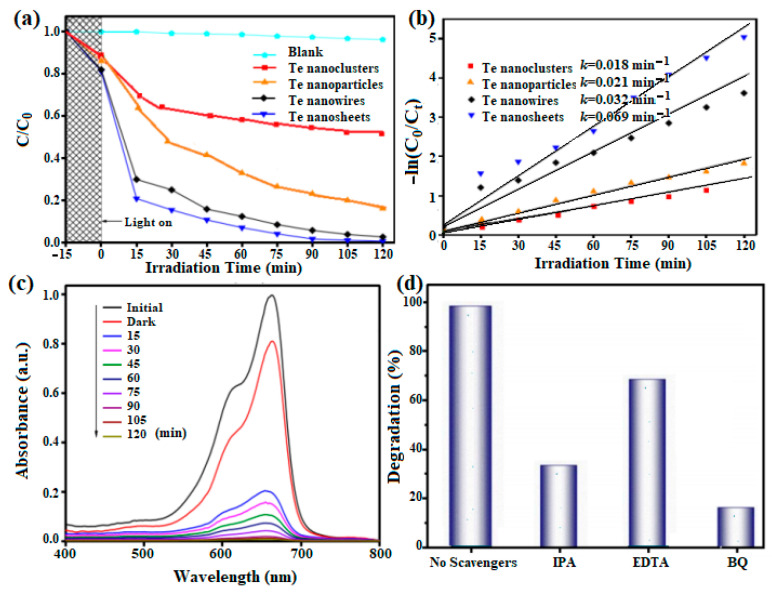
(**a**) Photodegradation efficiencies and: (**b**) photocatalytic kinetics of MB using Te with various nanostructures under visible light irradiation; (**c**) time-dependent absorption spectra for MB; and (**d**) effects of different scavengers on the photodegradation of MB using Te nanosheets.

**Figure 7 micromachines-15-00001-f007:**
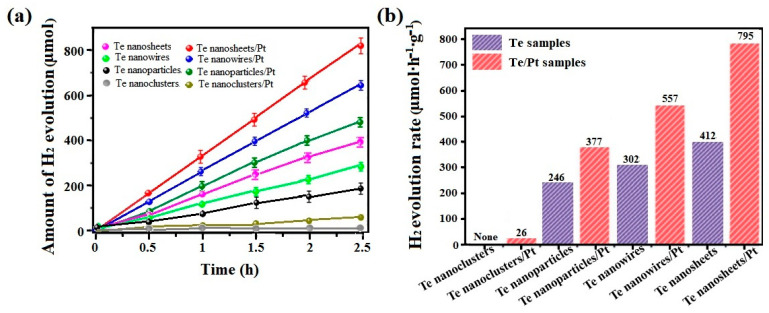
Visible light photocatalytic H_2_ evolution, determined using Te with various nanostructures and Te samples loaded with Pt. (**a**) Cumulative amount of H_2_ with time, and (**b**) H_2_ evolution rates.

**Figure 8 micromachines-15-00001-f008:**
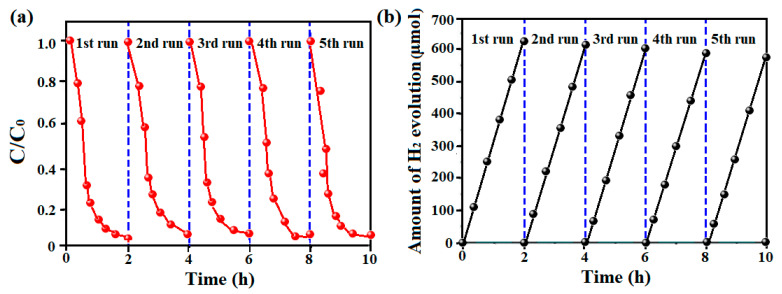
Stability test of (**a**) photodegradation and (**b**) photocatalytic H_2_ generation for MB over Te nanosheets under visible light illumination.

## Data Availability

The data presented in this study are available on request from the corresponding author.

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
