# Peer review of "Controllable Synthesis, Formation Mechanism, and Photocatalytic Activity of Tellurium with Various Nanostructures"

_micromachines, 2023, doi:10.3390/mi15010001_

Round 1

Reviewer 1 Report

Comments and Suggestions for Authors

Tellurium (Te) is a P-type semiconductor, in which its nanostructures all exhibit the photocatalytic activities for degradation of MB and H2 production under VL irradiation, especially Te nanosheets. It indicate that Te with various nanostructures are potential photocatalysts in the photocatalysis field, and this study is meaningful for visible light photocatalysis researches. However, some minor points should be addressed before accepting it for publication. 

(1) The nanostructured Tellurium (Te), as a P-type semiconductor, has potential application under VL in the field of photocatalysis, so please explain why 2D Te has better photocatalytic performance compared with other nanostructures?

(2) The structural comparison shows the shifting phenomena of peak positions, such as Raman and FTIR, please explain the reason for this phenomenon?

(3) The ordinate representation in Fig.3c is incomplete, please complete it.

(4) In the reference part, there is a format inconsistency, eg. upper and lower case, upper and lower corner mark, etc., please verify and correct.

Comments on the Quality of English Language

There are some minor writing and grammar errors, please read and correct them carefully.

Reviewer 2 Report

Comments and Suggestions for Authors

In this manuscript, the authors studied the synthesis of various Te nanostructures and their application in photocatalysis like degradation of dye and H2 production. I suggest to accept this manuscript after careful consideration on the following comments. Major modifications and significant supporting data are required before the resubmission.

Comments:

1. An important research on the synthesis and application of Te nanostructures is missing in the reference. Science Advances 2018, 4, eaas9894

2. The morphology characterization of Te nanostructures is far from enough. As the authors claimed in the paper, the morphology of Te nanostructures show different performances in photodegradation and H2 production. They should give the SEM results of large-area Te nanostructures (like 100 μm²), especially the 0D and 2D Te nanostructures.

3. In Fig. 1b, the authors should add the size distribution result of the uneven nanoparticles. In Fig. 1d, the author should measure the thickness of Te nanosheet. These results will influence the photocatalytic performance and their explanation on difference.

4. The inset image in Fig. 2d is not clear and its quality should be improved.

5. In Fig. 3, the authors have no description for the samples in UV-vis absorption test. Are they solid or in solution? What is the substrate? Do they have the same weight or concentration? Similarly, these important information is required in the photocatalysis tests.

Comments on the Quality of English Language

There are some grammatical errors:

  In the abstract section, the tense needs to be consistent. Like “…, which increased the output by nearly 2 times”

  In the Results and Discussion section, “Obviously, all diffraction peaks of Te products …. better crystallinity, which ....”

  What does TOC refer to ?

Reviewer 3 Report

Comments and Suggestions for Authors

This paper is fundamentally sound and well-written, and the results are interesting. But, there are some minor drawbacks. I will encourage the authors to revise the paper appropriately. 

p2, line 92 onwards, the texts should be carefully checked to avoid misunderstandings and repetitions. 

The well-accepted definition of 0D nanomaterials is either spherical or quasi-spherical materials that process a diameter of less than 100 nm. From the SEM images, quite many of the claimed 0D nanomoparticles, visually,  have a diameter of larger than 100 nm. Thus, please elaborate more on the statistical results.

Fig.1, what is the lateral thickness of the 2D sheets? 

Comments on the Quality of English Language

Overall, the quality of English is good. But, there is also an unfortunate lack of rigor in the writing. Some examples of sentences that need improving:

'Te nanostructures all have complete grain sizes with excellent crystallinity and mesopore structures.' complete grain sizes? 

'The presence of 1.0 g PVP results in the synthesis of clearer, uniform and straight nanowire with a diameter about of 25 nm are obtained'. 

'The recyclability of photocatalysts be crucial to its future practical applications.' 

etc. 

Round 2

Reviewer 2 Report

Comments and Suggestions for Authors

I can recommend it for publication.